# Improvement of Theory of Mind in Schizophrenia: A 15-Year Follow-Up Study

**Oguz Kelemen [1,2,\*], Adrienne Máttyássy [2] and Szabolcs Kéri [3,4,5]**

[1]    Department of Behavioral Sciences, University of Szeged, 6722 Szeged, Hungary
[2]    Bács-Kiskun County Hospital, Psychiatry Centre, 6000 Kecskemét, Hungary
[3]    Nyirő Gyula National Institute of Psychiatry and Addictions, 1135 Budapest, Hungary
[4]    Department of Cognitive Science, Budapest University of Technology and Economics, 1111 Budapest, Hungary
[5]    Department of Physiology, University of Szeged, 6722 Szeged, Hungary
\*    Correspondence: kelemen.oguz@med.u-szeged.hu; Tel.: +36-62-420-530

**Abstract:** Neurocognitive and social cognitive deficits are a hallmark of schizophrenia. The purpose of the present study was to investigate long-term changes in theory of mind (ToM), executive functions, lexical retrieval, and speed of information processing/attention in schizophrenia. We followed-up 31 outpatients with schizophrenia and 31 healthy control subjects for 15 years. ToM was assessed with the Reading the Mind from the Eyes Test (RMET), whereas neurocognitive functions were measured with the verbal fluency (VF) task (executive functions and lexical retrieval) and with the Digit-Symbol Substitution Test (DSST) (speed of information processing/attention). Clinical symptoms and general functioning were rated with the Positive and Negative Syndrome Scale (PANSS) and with the Global Assessment of Functioning (GAF) scale, respectively. At baseline assessment, patients with schizophrenia exhibited significant and generalized impairments on all measures. At follow-up, relative to the baseline, we observed marked improvements in ToM (RMET), stability in executive functions and lexical retrieval (VF), and a significant decline in psychomotor speed/attention (DSST) in schizophrenia. Clinical symptoms and psychosocial functions did not differ at baseline and at follow-up examinations (mild-to-moderate symptoms on the PANSS and moderate difficulty in social and occupational functions on the GAF). These results indicate that patients with schizophrenia with mild-to-moderate symptoms and functional deficits are characterized by improved ToM during over a decade.

**Keywords:** schizophrenia; social cognition; Reading the Mind from the Eyes Test; endophenotype; follow-up study; neurocognitive dysfunction

## 1. Introduction

Theory of mind (ToM) refers to our ability to understand the internal states of other individuals, including the representation of intentions and complex social emotions. ToM dysfunctions, as characterized by laboratory tests, are thought to be a hallmark feature of schizophrenia [1–5], which is present before the onset of psychosis [6,7] and in the biological relatives of schizophrenia patients [8–10]. Therefore, ToM dysfunctions may be viewed as potential endophenotypes [11–14], although the results are not equivocal and straightforward [15–20]. On the other hand, several studies highlighted the relevance of ToM functions in clinical outcome, functional prognosis, and quality of life in schizophrenia [21,22].

Long-term follow-up studies are essential to better understand the prognosis and outcome of schizophrenia and to identify potential state-independent endophenotypes. By reviewing the literature,

Jobe and Harrow [23] concluded that patients with schizophrenia regularly showed poorer functional outcomes than patients with other psychotic disorders. However, some patients had extended periods of recovery, and, contrary to the commonly held view, patients with schizophrenia were not characterized by a progressive decline. More recently, Volavka and Vevera [24] concluded that symptomatic remission in long-term follow-up studies ranged between 16.4% in never-treated patients to 37.5% in patients who received antipsychotics. Better outcomes were associated with early detection and treatment of the first episode, and continuous psychosocial support over subsequent years [24].

However, there is a paucity of follow-up studies focusing on ToM and neurocognitive dysfunctions in schizophrenia. Cook et al. [25] investigated ToM by using the hinting test in a one-year follow-up study. In a sample of 43 outpatients with severe mental illness (schizophrenia and schizoaffective disorder), the authors found that neurocognitive improvement was correlated with better social functions, but, contrary to the expectations, the performance on the ToM task did not predict social functions [25].

In a two-year follow-up study, Piskulic et al. [7] assessed ToM with the social inference subscale of the Awareness of Social Inference Test, together with the examination of facial emotion perception (Penn Emotion Recognition and Differentiation tasks), in individuals at high-risk for psychosis. Their results indicate a mild-to-moderate and continuous impairment in ToM, suggesting that it is a vulnerability factor for schizophrenia. However, there was no relationship between ToM impairments and conversion to psychosis [7].

Ayesa-Arriola et al. [26] applied the Reading the Mind from the Eyes Test (RMET) in a 3-year follow-up study. From the first psychotic episode, they investigated 160 patients three times (baseline, one year, three years). The data of Ayesa-Arriola et al. [26] indicated that the ToM deficit was a stable, trait-like impairment, which did not change during the follow-up period. ToM deficits were not associated with clinical symptoms, but there was a relationship between ToM and neurocognitive dysfunctions, especially in relation to processing speed.

Mc Clerry et al. [27] examined social cognitive abilities in 41 patients with schizophrenia in a 5-year follow-up study. Using a self-report emotional intelligence test (Mayer-Salovey-Caruso Emotional Intelligence Test) and the Relationships Across Domains test for social perception, they found a stable performance, which supports the assumption that social cognitive impairment may be a trait marker of schizophrenia. Martino et al. [28] found similar stability in basic emotion recognition among patients with bipolar disorder (seven-year follow-up).

The aim of the present study was to investigate how ToM may change in schizophrenia over a decade. To our knowledge, there are no similar long-term investigations into ToM functions in schizophrenia. Based on previous results [11–14], we hypothesized that patients with schizophrenia will show similar or worse performances on the RMET at the follow-up testing session relative to the baseline values. In addition, we also used tests for executive functions, attention, and speed of information processing to compare social cognitive (RMET) and neurocognitive functions.

## 2. Methods

### 2.1. Participants

The patients were recruited at the Bács-Kiskun Country Hospital (Kecskemét, Hungary) and at the University of Szeged (Szeged, Hungary) (initial recruitment: 1999–2002). The diagnosis was established according to the DSM-IV criteria [29] by using the MINI International Neuropsychiatric Interview [30]. Baseline data from some of our participants were previously published [31,32]. After more than 10 years, 31 out of the original 41 patients were able to participate (retest: 2016–2018) who regularly visited mental health care facilities. There was no evidence for mortality in the remaining 10 patients who were not included in the follow-up assessment. Each patient was assessed with the Positive and Negative Syndrome Scale (PANSS) [33]. General psychosocial functions were evaluated with the Global Assessment of Functioning (GAF) scale [29]. All patients received antipsychotic medications

(flupenthixol, zuclopenthixol, olanzapine, quetiapine, risperidone, paliperidone, amisulpride). The clinical and demographic parameters are presented in Table 1. The healthy control subjects, matched for age, gender, and education to the schizophrenia patients, were hospital employees and their relatives, and acquaintances. The study was carried out in accordance with the Declaration of Helsinki and was approved by the institutional ethics board (University of Szeged). All subjects gave their informed consent.

**Table 1.** Clinical and demographic parameters of the participants.

|  | Schizophrenia (n = 31) | Control Subjects (n = 31) |
|---|---|---|
| Age at baseline (years) | 34.2 (8.4) | 35.1 (7.5) |
| Male/female | 19/12 | 19/12 |
| Education (years) | 11.6 (4.3) | 11.5 (4.6) |
| Employment rate (baseline-follow-up) | 17 (55%)–19 (61%) | 23 (74%)–23 (74%) |
| Duration of illness at baseline (years) | 7.9 (3.4) | - |
| PANSS (baseline) | P: 10.3 (4.0) N: 12.7 (5.1) G: 47.3 (9.5) | |
| PANSS (follow-up) | P: 9.5 (4.5) N: 13.5 (5.8) G: 49.4 (10.3) | - |
| GAF (baseline) GAF (follow-up) | 57.4 (10.2) 56.5 (9.7) | - |
| CPZ (baseline) CPZ (follow-up) | 435.9 (147.0) 446.8 (191.5) | - |
| Proportion of second-generation antipsychotics Baseline Follow-up | 35 76% | - |

Data are mean (standard deviation). PANSS—Positive and Negative Syndrome Scale, P—positive, N—negative, G—general, GAF—Global Assessment of Functioning, CPZ—chlorpromazine-equivalent dose of antipsychotics (mg/day). There was no significant difference between the two groups in age, education, and gender ratio ($p$s > 0.2). PANSS, GAF, and CPZ values at baseline and follow-up did not differ significantly ($p$s > 0.2).

## 2.2. Assessment of Social Cognition and Neurocognition

In the revised version of the RMET, 36 photographs of the eye-region of faces, expressing complex social emotions, were presented. Without any time-pressure, subjects were asked to choose which of four words (one target and three foils) best described the mental state of the actor/actress on the picture [34].

We assessed neurocognitive functions using the verbal fluency (VF) test (Controlled Oral Word Association Test) [35] and Digit Symbol Substitution Test (DSST) [36]. These tests provide reliable information on executive functions, attention, and speed of information processing, which are key indices of cognitive impairments in schizophrenia [37,38], and their adapted forms were previously used in Hungarian populations [15,31,32].

## 2.3. Data Analysis

STATISTICA 13 (Tibco) software was used for data analysis. The raw data were *z*-transformed. Repeated-measures analyses of variance (ANOVA) were used to compare patients and controls for RMET, VF, and DSST and to investigate the effect of assessment (baseline vs. follow-up). In this ANOVA, the between-subjects factor was the experimental group (schizophrenia vs. controls), and the within-subjects factors were the time of assessment (baseline vs. follow-up) and test type (RMET, VF, and DSST). Tukey's honestly significant difference (HSD) tests were used for post-hoc comparisons.

Clinical and demographic variables were compared with two-tailed Student's *t*-tests. Pearson's product moment correlation coefficients were calculated between clinical measures, RMET, VF, and DSST. The level of statistical significance was set at alpha < 0.05. The false discovery rate (FDR) method was used for the correction of multiple comparisons.

## 3. Results

### 3.1. Differences between Patients with Schizophrenia and Healthy Controls at Baseline and Follow-up

First, the raw data (Table 2) were *z*-transformed in order to compare different test results in the same analysis. The results of the main ANOVA are summarized in Table 3. The three-way interaction among group, assessment, and test type indicated a significant difference between patients with schizophrenia and healthy controls, which exhibited a distinct pattern at baseline and follow-up assessment.

**Table 2.** Raw data from the neurocognitive and social cognitive measures

|  | Schizophrenia (n = 31) | Control Subjects (n = 31) |
|---|---|---|
| **Verbal fluency (VF)** | | |
| Baseline | 35.8 (12.4) | 42.9 (11.3) |
| Follow-up | 36.9 (12.1) | 42.5 (11.2) |
| **Digit-Symbol Substitution Test (DSST)** | | |
| Baseline | 56.8 (16.0) | 68.4 (13.5) |
| Follow-up | 47.7 (16.3) | 68.2 (14.1) |
| **Reading the Mind from the Eyes (RMET)** | | |
| Baseline | 19.6 (11.7) | 26.1 (9.7) |
| Follow-up | 23.8 (10.2) | 25.9 (9.3) |

Data are mean (standard deviation). The measures were as follows: VF—number of words beginning with letters F, A, and S in 60 s; DSST—number of empty boxes completed in 90 s; RMET—number of correctly identified mental states.

**Table 3.** Results from the analysis of variance (ANOVA) comparing patients with schizophrenia and healthy control subjects at baseline and follow-up assessments on the three tests.

| Main Effect or Interaction | *F* | *df* | *p* | $\eta^2$ |
|---|---|---|---|---|
| Group (schizophrenia vs. controls) | 70.77 | 1, 60 | <0.001 | 0.54 |
| Assessment | 0.21 | 1, 60 | 0.64 | 0.004 |
| Test type | 10.77 | 2, 120 | <0.001 | 0.15 |
| Assessment by group | 0.002 | 1, 60 | 0.96 | <0.001 |
| Test type by group | 9.99 | 2, 120 | <0.001 | 0.14 |
| Assessment by test type | 18.21 | 2, 120 | <0.001 | 0.23 |
| Assessment by test type by group | 18.44 | 2, 120 | <0.001 | 0.24 |

The ANOVA was performed on the z-standardized scores from three types of tests: reading the mind from the eyes test (RMET) (theory of mind), verbal fluency (executive functions and lexical retrieval), and digit-symbol substitution test (speed of information processing, attention, and motor skills). At baseline, patients with schizophrenia were significantly impaired on the RMET, VF, and DSST relative to the control subjects (*p*s < 0.005). At follow-up assessment, we observed significant differences between patients and controls for VF and DSST (*p*s < 0.005), but not for RMET (*p* > 0.5) (Figure 1). We found a highly significant improvement in RMET performances in schizophrenia (baseline vs. follow-up, *p* < 0.001), no changes in VF scores (*p* > 0.5), and a significant decline in DSST (*p* < 0.001). There were no significant correlations between PANSS scores, GAF, chlorpromazine-equivalent antipsychotic doses, duration of illness, and test performances (−0.2 < *r*s < 0.2, *p* > 0.1).

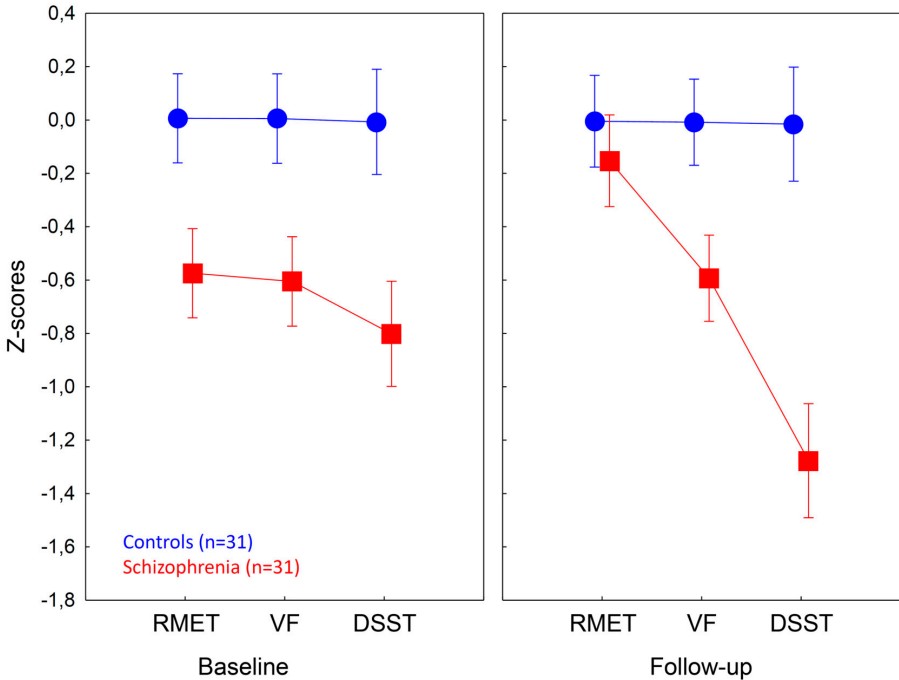

**Figure 1.** Comparison of patients with schizophrenia and healthy control subjects on the social cognitive and neurocognitive tests. RMET—Reading the Mind from the Eyes Test, VF—Verbal Fluency, DSST—Digit-Symbol Substitution Test (DSST). At baseline, patients with schizophrenia were impaired on all measures ($p$s < 0.001, Tukey's HSD tests), whereas at follow-up, significant between-group difference was not observed for RMET. During the follow-up period, patients with schizophrenia showed significant improvements for RMET and worsening for DSST ($p$s < 0.001). Error bars indicate 95% confidence intervals.

## 3.2. Control Analyses

We also performed a series of control analyses to investigate the test-retest properties of the cognitive tasks and their relationships. First, in the control group, RMET, VF, and DSST scores showed a remarkable stability over time (Figure 1) (baseline vs. follow-up comparisons, $p$s > 0.5). We also observed high test-retest correlations for RMET ($r = 0.73$, $p < 0.001$), VF ($r = 0.78$, $p < 0.001$), and DSST ($r = 0.79$, $p < 0.001$) in the control group. In the schizophrenia group, as expected, there was no significant correlation between baseline and follow-up RMET scores ($r = 0.29$, $p = 0.11$). However, we found significant correlations between baseline and follow-up VF scores ($r = 0.8$ $p < 0.001$) and DSST scores ($r = 0.74$, $p < 0.001$) in patients with schizophrenia. Correlation analyses in the whole sample confirmed that VF and DSST tap a similar underlying cognitive construct (correlations between VF and DSST scores: $r = 0.85$, $p < 0.001$), meanwhile RMET and VF ($r = 0.19$, $p = 0.12$) and RMET and DSST ($r = 0.21$, $p = 0.10$) does not.

## 4. Discussion

Contrary to our initial expectations, the present data suggest that ToM, as measured by the RMET, improve in over a decade in schizophrenia. Therefore, ToM functions are less likely to be considered as a trait marker of schizophrenia. The findings of our study seem to stand at odds with the results of Ayesa-Arriola et al. [26] who found stable RMET performances in psychosis over a 3-year follow-up period. There are several differences between the studies that can explain the seemingly discrepant findings. First, in our study, the follow-up period was longer, but the sample size was smaller as compared to the study of Ayesa-Arriola et al. [26]. Second, Ayesa-Arriola et al. [26] included first-episode patients with psychosis, whereas our sample consisted of more chronic cases

with schizophrenia. Third, an important selection bias is very likely in our study. Specifically, over a decade-long follow-up period, only clinically stable patients with good adherence and functional outcome could be retained in a natural clinical setting. Indeed, our patients were characterized by moderate symptoms and some difficulty in social and occupational functioning, as indicated by the PANSS and the GAF scores, and all of them lived independently in the community. The GAF scores indicated moderate difficulty in social and occupational functioning (e.g., few friends, conflicts with peers or co-workers) [29]. The employment rate in the schizophrenia group was unusually high (55–61%) as compared to data from other European countries (12–39%) [39]. Therefore, the finding of improved ToM in later phases of schizophrenia might be generalized only to schizophrenia patients with mild-to-moderate symptoms, moderate functional deficits, and good treatment adherence.

The mechanism and relevance of improved ToM are not clear, because the RMET scores were not associated with clinical symptoms and social functions, which were very similar at baseline and follow-up assessment. An obvious explanation is that a more widespread application of second-generation antipsychotics may lead to better social cognition. However, the literature does not uniformly support this assumption [40]. In addition, the potential superiority of newer antipsychotics in cognitive enhancement was not supported by the declining DSST performance in our sample. It is notable that most of our patients participated in regular supportive therapy and remediation programs. It is possible that intense social interactions may contribute to enhanced ToM.

It is particularly interesting that VF was stable, whereas the DSST performance declined in patients with schizophrenia. A persistent and non-progressive VF impairment has widely been documented in the literature [41,42]. According to the meta-analysis of Szöke et al. [42], semantic verbal fluency is the most likely trait marker of schizophrenia, which does not change in follow-up studies. In contrast to VF, the DSST measures speed of information processing and attention, but the contribution of working memory and executive functions is not negligible. The DSST is very sensitive but less specific to a wide range of impairments and is especially suitable for the detection of longitudinal cognitive changes in psychiatric disorders [43]. The dissociation among RMET (improved), VF (stable), and DSST (worsened) is surprising, suggesting that distinct cognitive domains display an altered pattern of changes during the course schizophrenia. One of the most important distinctions was between social cognition (RMET) and neurocognition (VF and DSST). These domains exhibited a very low correlation in our sample, which is consistent with earlier conclusions [4,44,45].

The present study is not without limitations: the sample size was small, and only a few cognitive functions were assessed by a limited number of tests in a natural clinical setting. Only the GAF scale was available for the characterization of psychosocial functions. However, evidence supports the appropriateness of social cognitive and neurocognitive tests used in our study [41,42,46–49]. The DSST is one of the best measures of general cognitive dysfunctions in schizophrenia. In a meta-analysis, Dickinson et al. [46] found that the weighted mean effect size for the DSST was higher than that for other measures of episodic memory, executive functioning, and working memory. The authors concluded that the DSST is reliable and clinically suitable measure of damaged information processing, which is a core feature of schizophrenia-associated cognitive dysfunctions [46]. Given that impaired DSST is the single most robust indicator in schizophrenia [46], the non-specific effect of illness burden and the adverse effects of long-term medication can be detected in the form of declined psychomotor speed. However, given the low specificity of the DSST, we cannot delineate a specific cognitive mechanism that exhibited a marked decline during the follow-up period. This question should be addressed by future studies with a more detailed neuropsychological assessment.

Recently, the psychometric properties of several social cognitive tasks were investigated in patients with schizophrenia [47–49]. In a two–week follow-up study, RMET performances showed an acceptable test-retest reliability [49], although it was not confirmed in an Asian population [47]. It is important to underline that each task used in our study yielded good test-retest properties, and the marked and opposite changes on RMET and DSST in schizophrenia can hardly be attributed to weak psychometric properties.

In conclusion, we found markedly improved social cognitive functions and declining speed of information processing in schizophrenia during a 15-year follow-up period. Future studies are warranted to clarify the neuronal correlates and clinical relevance of these long-term cognitive changes.

**Author Contributions:** O.K. and S.K. designed and planned the experiments. O.K., A.M., and S.K. carried out the experiments, and contributed to the interpretation of the results; were involved in drafting the article or revising it critically, provided final approval of the, version to be published; and agreed to be accountable for all, aspects of the work in ensuring that questions related to the accuracy or integrity of any part of the work were appropriately investigated and resolved.

**Funding:** This work was supported by the BME-Biotechnology FIKP grant of EMMI (BME FIKP-BIO), and by the National Research, Development and Innovation Office (NKFIH K18/K128559).

**Acknowledgments:** We would like to acknowledge the help of Rita Erdélyi for recruiting and scoring the patients.

**Conflicts of Interest:** The authors declare no conflict of interest.

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
