# Peer review of "Improvement of Theory of Mind in Schizophrenia: A 15-Year Follow-Up Study"

_psych, doi:10.3390/psych1010032_

Round 1

Reviewer 1 Report

The paper reports results from a more than 10 years long follow-up study on how a group of 31 patients with schizophrenia performs on neurocognitive and theory-of-mind measures compared to healthy control participants. Contrary to expectation, the patients improve their scores on the ToM measure, indicating that this aspect of social cognition should not be listed as a trait measure of schizophrenia. They authors also report a worsening of psychomotor speed.

·       The paper is well written.

·       The introduction shortly summarizes findings and current aim.

·       I was expecting to find the method chapter with details of participants and methods, before going on to the results. I assume the Journal has asked for this format, but I find it awkward to go back and forth like I had to do while reading.

·       No explicit references to the two neurocognitive tests were given, only to other papers by the authors where the methods had previously been used.

·       Besides, all methods were originally made for English speaking subjects. How this was handled in a non-English speaking study, is not discussed.

·       I have no objections to the ANOVA being conducted on z-scores, but I miss details on the actual performances on the tests analyzed.  

·       Also, I would prefer that the control group’s baseline scores were used for computing the z-scores, so that also the change in the control group’s performance could be visible. Actually, the psychometric values could be presented as footnote to a table showing the test results at baseline and follow-up for the two groups (raw scores), accompanied with the figure of the z-sores.

·       I agree with the authors that the three-way interaction is what matters the most, and which needs to be followed up by post-hoc pair-wise comparisons as done.

·       In the Discussion, the authors list shortcomings of the sample and say that only patients with good adherence and thus good outcome were available, which may explain why the results differ from others. The definition of good outcome is a tricky one and should be qualified. In any case, I find this disqualification somewhat awkward since status of the sample was known beforehand and could be listed as an asset of the study by re-framing the hypothesis and thus strengthen a more optimistic view of schizophrenia outcome with the current findings.

·       Lastly, finding that DSST-scores decline could be in line with the more general finding reported by Dickinson et al in 2007 that impairments on this test is the single most robust finding in schizophrenia. Being ill for > 10 years and being on medication, probably affect the nervous system in some way or another. The paper could point to the possibility of declined psychomotor speed being a sensitive measure of the cost.

·       There is an unnecessary sentence on line 212 (page 7) in the reference chapter.

·       In the abstract, start by stating that the purpose of the study is to investigate the domains, not the measures, and then specify which measures are being used. Also, stick to the same when summing up the results. The last statement, informing the reader that the patient group has ‘good prognosis and outcome’, is new to the reader. Some qualifying information should be given in the abstract. The information given tells us that the group is stable in symptoms and functioning. That is not necessarily the same as good prognosis & outcome.

Author Response

We are indebted for the insightful comments and constructive criticism. We modified the manuscript following these comments and requests. Each query (Q) is cited below together with the responses (R). In the paper, changes are shown in red. In our responses, we refer to the page number (p) and the number of lines (ln) where the changes can be found.

Q1. “I was expecting to find the method chapter with details of participants and methods, before going on to the results. I assume the Journal has asked for this format, but I find it awkward to go back and forth like I had to do while reading.

R1: Thank you for this important note. We modified the format of the paper according to the classic, clearer schema (introduction – methods – results – discussion).

Q2: “No explicit references to the two neurocognitive tests were given, only to other papers by the authors where the methods had previously been used.

R2: In the revised paper, we referred to the original sources of the tests, and we highlighted our previous papers once again where these methods were used in Hungarian populations (p. 3, ln. 110-114).

Q3: “Besides, all methods were originally made for English speaking subjects. How this was handled in a non-English speaking study, is not discussed.

R3: The methods were adapted for Hungarian populations, and we used them in our previous work. Now, this work is highlighted and clearly referred (see R2).

Q4: “I have no objections to the ANOVA being conducted on z-scores, but I miss details on the actual performances on the tests analyzed. Also, I would prefer that the control group’s baseline scores were used for computing the z-scores, so that also the change in the control group’s performance could be visible. Actually, the psychometric values could be presented as footnote to a table showing the test results at baseline and follow-up for the two groups (raw scores), accompanied with the figure of the z-sores.”

R4: We added a new table (Table 2) to the paper in which the raw scores are summarized. We believe that it is more visible than a footnote. From this table it is evident that the scores were quite stable in the control group (p. 4.).

Q5: “In the Discussion, the authors list shortcomings of the sample and say that only patients with good adherence and thus good outcome were available, which may explain why the results differ from others. The definition of good outcome is a tricky one and should be qualified. In any case, I find this disqualification somewhat awkward since status of the sample was known beforehand and could be listed as an asset of the study by re-framing the hypothesis and thus strengthen a more optimistic view of schizophrenia outcome with the current findings.

R5: We did not include specific measures of outcome in the present study (e.g. the Personal and Social Performance (PSP) Scale or the UCSD Performance-based Skills Assessment (UPSA)). This is a very important but ambiguous research field given that there is no consensus how to measure outcome and prognosis. For example, Isaac et al. (Br J Psychiatry Suppl. 2007;50:s71-7) found that.the most frequent indicators of outcome are clinical symptoms, hospitalization and mortality (direct indicators), and social/occupational functioning, marriage, social support and burden of care (indirect indicators). In the present study, indirect information can be obtained from the GAF scores, and from the fact that all patients lived in the community independently. In addition, we added the employment rate of the patients to the paper, which was unusually high. Moderate psychosocial problems, high employment rate, mild symptoms, and living independently in the community all indicated good outcome (p. 3., Table 1; p. 6., ln. 191-197).

Q6: “Lastly, finding that DSST-scores decline could be in line with the more general finding reported by Dickinson et al in 2007 that impairments on this test is the single most robust finding in schizophrenia. Being ill for > 10 years and being on medication, probably affect the nervous system in some way or another. The paper could point to the possibility of declined psychomotor speed being a sensitive measure of the cost.

R6: This point of view has been added to the discussion (p. 7, ln. 227-230).

Q7: “There is an unnecessary sentence on line 212 (page 7) in the reference chapter.

R7: The unnecessary sentence has been deleted.

Q8: “In the abstract, start by stating that the purpose of the study is to investigate the domains, not the measures, and then specify which measures are being used. Also, stick to the same when summing up the results. The last statement, informing the reader that the patient group has ‘good prognosis and outcome’, is new to the reader. Some qualifying information should be given in the abstract. The information given tells us that the group is stable in symptoms and functioning. That is not necessarily the same as good prognosis & outcome.

R8: We modified the abstract following the instructions (p. 1). We defined domains and measures separately, and we attempted to make a distinction between symptoms/functioning (directly quantified in the study) and outcome/prognosis (not directly and specifically assessed in the study) in the whole manuscript (see also R5).

Reviewer 2 Report

Original study which emphasizes the need for research without short term grants.

This is a very interesting long term study in neurocognitive and social congitive funtioning that one does not come across easily.

I have some small comments:

Long term follow up studies in schizophrenia have been done, but not focussed on cognition or theory of mind. One or two of such studies can be inserted as contrast. 

The name and function of the figure on page 2 is unclear: it is not referred to in the text and its value for the article is unclear. Please clarify this figure, add a name to it, include it in the text and comment on the outcome or leave it out of the manuscript.

Duration of disease is not mentuioned or analysed. Please add this to the demographic data and results. 

Mortality is increased in schizophrenia. 15 year follow up is a period of time in which mortality could have occurred. Please mention if it did occur and, if so, imentuion the number of patients and describe the cause(s) of death.

Author Response

We are indebted for the insightful comments and constructive criticism. We modified the manuscript following these comments and requests. Each query (Q) is cited below together with the responses (R). In the paper, changes are shown in red. In our responses, we refer to the page number (p) and the number of lines (ln) where the changes can be found.

Q1: “Long term follow up studies in schizophrenia have been done, but not focussed on cognition or theory of mind. One or two of such studies can be inserted as contrast.”

R1: We added two key reviews of such studies in the Introduction (p. 1-2, ln. 42-51).

Q2: “The name and function of the figure on page 2 is unclear: it is not referred to in the text and its value for the article is unclear. Please clarify this figure, add a name to it, include it in the text and comment on the outcome or leave it out of the manuscript.

R2: We appropriately inserted a number and legend attached to the figure, which is also referred in the text (p5., Fig. 1).

Q3: “Duration of disease is not mentuioned or analysed. Please add this to the demographic data and results.”

R3: We added the duration of illness to the manuscript and included this parameter in the analysis. The duration of illness does not correlate with test performances (p. 3., Table 1; p. 5., ln. 155).

Q4: “Mortality is increased in schizophrenia. 15 year follow up is a period of time in which mortality could have occurred. Please mention if it did occur and, if so, imentuion the number of patients and describe the cause(s) of death.”

R4: There was no evidence for mortality in the remaining 10 patients who were not followed-up in cognitive assessment (p. 2., ln. 90-91).